# Necessary and Sufficient Watermark for Large Language Models

## Abstract

In recent years, large language models (LLMs) have achieved remarkable performances in various NLP tasks. They can generate texts that are indistinguishable from those written by humans. Such remarkable performance of LLMs increases their risk of being used for malicious purposes, such as generating fake news articles. Therefore, it is necessary to develop methods for distinguishing texts written by LLMs from those written by humans. Watermarking is one of the most powerful methods for achieving this. Although existing watermarking methods have successfully detected texts generated by LLMs, they significantly degrade the quality of the generated texts. In this study, we propose the Necessary and Sufficient Watermark (NS-Watermark) for inserting watermarks into generated texts without degrading the text quality. More specifically, we derive minimum constraints required to be imposed on the generated texts to distinguish whether LLMs or humans write the texts. Then, we formulate the NS-Watermark as a constrained optimization problem and propose an efficient algorithm to solve it. Through the experiments, we demonstrate that the NS-Watermark can generate more natural texts than existing watermarking methods and distinguish more accurately between texts written by LLMs and those written by humans. Especially in machine translation tasks, the NS-Watermark can outperform the existing watermarking method by up to 30 BLEU scores.

## 1 Introduction

Large language models (LLMs) have achieved remarkable performances in a wide range of NLP tasks, including language generation (Chen et al., 2021), question answering (Joshi et al., 2017; Kwiatkowski et al., 2019), and reasoning tasks (Bisk et al., 2020; Kojima et al., 2022). Recently, many pre-trained LLMs have been released (Brown et al., 2020; Chung et al., 2022; Zhang et al., 2022; Touvron et al., 2023), which can now generate natural and fluent texts that are indistinguishable from texts written by humans. For instance, Brown et al. (2020) evaluated the quality of the news articles generated by GPT-3, demonstrating that humans cannot distinguish between news articles generated by GPT-3 and those written by humans.

As the performance of LLMs improves for various tasks, the risk that LLMs are used for malicious purposes, such as generating fake news, also increases (Zellers et al., 2019). Thus, it is crucial to develop methods to identify whether LLMs or humans write texts. Watermarking is one of the powerful techniques for this purpose, which inserts information into texts such that the inserted information is imperceptible to humans and can be easily identified by some algorithms (Venugopal et al., 2011; He et al., 2021; 2022; Zhao et al., 2023b; Kuditipudi et al., 2023; Zhao et al., 2023a; Kirchenbauer et al., 2023a;b; Christ et al., 2023). Recently, Kirchenbauer et al. (2023a) proposed the Hard/Soft-Watermark,[1] which inserts watermarks by generating text using only a subset of vocabulary. Texts generated by LLMs consist only of a subset of vocabulary, whereas texts written by humans consist of an entire vocabulary. Thus, we can identify whether LLMs or humans write text using statistical hypothesis testing. Kirchenbauer et al. (2023a) demonstrated that the Hard/Soft-Watermark can identify whether LLMs or humans write texts almost perfectly. However, because LLMs generate texts with only a subset of vocabulary, the generated texts are often of low quality.

---

[1] We coined the name "Hard/Soft-Watermark" in this work to refer to the watermarking methods by Kirchenbauer et al. (2023a).

In this study, we propose a novel method for inserting watermarks into generated text without sacrificing both text quality and detection accuracy, which we refer to as the **Necessary and Sufficient Watermark (NS-Watermark)**. Our method is based on the observation that the constraint imposed by the Hard/Soft-Watermark is overly conservative for identifying LLM-generated texts, especially when the generated texts are long. Hence, we derive minimum constraints required to be imposed on the generated texts to detect LLM-generated texts. We find that the constraints on the generated text can be relaxed without decreasing the detection accuracy as the length of the generated text increases. Based on this observation, we propose the NS-Watermark, which can change the constraints according to the length and impose minimum constraints on the generated text. Owning to the minimum constraints, the text generated with the NS-Watermark can be more natural than the text with the Hard/Soft-Watermark. We experimentally evaluate the effectiveness of the NS-Watermark and demonstrate that the NS-Watermark can outperform the Soft-Watermark in terms of both text quality and detection accuracy. Particularly in the machine translation tasks, we demonstrate that the NS-Watermark can outperform the Soft-Watermark by up to 30 BLEU scores and achieve competitive BLEU scores compared to conventional decoding methods without watermarks.

## 2 BACKGROUND

In this section, we briefly describe the watermarking methods proposed by Kirchenbauer et al. (2023a). Further discussions on related studies are deferred to Sec. 6.

**Hard-Watermark.** Let $x_{\text{prompt}}$ be a prompt, $V$ be vocabulary, and $\gamma \in (0, 1)$ be a hyperparameter. Given a word $x_t$, using $x_t$ as the seed value, we randomly split $V$ into two disjoint subsets: *green words* $V^{\text{green}}(x_t)$ and *red words* $V^{\text{red}}(x_t)(:= V \setminus V^{\text{green}}(x_t))$ such that $|V^{\text{green}}(x_t)| = \gamma|V|$. Then, the Hard-Watermark generates text as follows:

$$\underset{x_{1:T},T}{\arg\max} \, p(x_{1:T} \mid x_{\text{prompt}}) \ \text{s.t.} \ x_{t+1} \in V^{\text{green}}(x_t) \ \ (t = 1, 2, \cdots, T-1). \tag{1}$$

If humans write the text, the green words appear with probability $\gamma$, whereas texts written by LLMs consist of only green words. Thus, using statistical hypothesis testing, we can identify whether LLMs or humans write the text. Specifically, the null and alternative hypotheses are given as follows:

$H_0$ : The green words appear in a text with probability $\gamma$.

$H_1$ : The green words appear in a text with probability greater than $\gamma$.

If we reject the null hypothesis, we can conclude that the text is generated by LLMs. The number of green words follows a binomial distribution in texts written by humans. Thus, we can test this by checking whether the z-score of text $x_{1:T}$, defined below, exceeds a given threshold $Z$.

$$z(x_{1:T}) := \frac{|x_{1:T}|_{\text{G}} - \gamma(T-1)}{\sqrt{\gamma(1-\gamma)(T-1)}}, \quad |x_{1:T}|_{\text{G}} := |\{x_{t+1} \mid x_{t+1} \in V^{\text{green}}(x_t)\}|. \tag{2}$$

**Soft-Watermark.** Although the Hard-Watermark is a simple and efficient method for distinguishing the texts written by LLMs from those written by humans, the generated texts are often of low quality. One reason for this is that the constraints of the Hard-Watermark may prevent the generation of common phrases, e.g., "Barack Obama," even if the probabilities of these phrases are very high because "Obama" may not be contained in $V^{\text{green}}$("Barack"). To mitigate this issue, Kirchenbauer et al. (2023a) also proposed the Soft-Watermark: instead of making all words contained in the generated text green words, the Soft-Watermark adds an offset and increases the probability of generating green words. This relaxation allows the Soft-Watermark to generate "Barack Obama" when the probability that "Obama" appears after "Barack" is high, and the Soft-Watermark can generate higher-quality text than the Hard-Watermark. However, the Soft-Watermark still suffers from low-quality text, as we demonstrate in the experiments.

## 3 PROPOSED METHOD

### 3.1 NECESSARY AND SUFFICIENT WATERMARK

In this section, we show that the constraints of the Hard/Soft-Watermark are too restrictive and derive the minimum constraint to identify whether LLMs or humans write the text. Rewriting Eq. (1), the

Hard-Watermark is reformulated as follows:

$$\underset{x_{1:T}, T}{\arg\max}\, p(x_{1:T} \mid x_{\text{prompt}}) \;\; \text{s.t.} \;\; \frac{|x_{1:T}|_{\text{G}}}{T-1} = 1. \tag{3}$$

Let $\hat{x}_{1:T}$ be the solution of Eq. (3). The z-score $z(\hat{x}_{1:T})$ is $\mathcal{O}(\sqrt{T})$, and we can identify whether LLMs or humans write the text by testing whether the z-score exceeds the hyperparameter $Z$. However, the z-score $z(\hat{x}_{1:T})$ increases with the length of the generated text $T$, whereas the threshold $Z$ remains constant. Therefore, the above formulation in Eq. (3) imposes too restrictive constraint on the generated text, especially when ensuring $z(x_{1:T}) \geq Z$ for long texts. More specifically, the following constraint is sufficient to ensure that the z-score of the generated text is greater than or equal to the threshold $Z$:

$$\underset{x_{1:T}, T}{\arg\max}\, p(x_{1:T} \mid x_{\text{prompt}}) \;\; \text{s.t.} \;\; \frac{|x_{1:T}|_{\text{G}}}{T-1} \geq \gamma + Z\sqrt{\frac{\gamma(1-\gamma)}{T-1}}. \tag{4}$$

If text $x_{1:T}$ is written by humans, the proportion of green words contained in a text $\frac{|x_{1:T}|_{\text{G}}}{T-1}$ is $\gamma$ on average. Thus, the second term in the constraint is the minimum margin for identifying whether the texts are written by LLMs. We refer to the above problem as the **Necessary and Sufficient Watermark (NS-Watermark)**. By comparing Eq. (3) with Eq. (4), the constraint of the NS-Watermark is looser than that in Eq. (3), although the z-score of the generated text is guaranteed to be greater than or equal to $Z$ because of the constraint in Eq. (4). Thus, the NS-Watermark can generate higher quality and more natural texts than the Hard/Soft-Watermark without decreasing detection accuracy. In the next section, we propose an efficient algorithm for computing the NS-Watermark.

### 3.2 NAIVE ALGORITHM FOR NECESSARY AND SUFFICIENT WATERMARK

The Hard/Soft-Watermark can be computed using the conventional beam search because the Hard-Watermark generates texts using only green words, and the Soft-Watermark just adds an offset to the probability that green words appear. However, the NS-Watermark needs to control the proportion of green words contained in generated texts and needs to optimize where green words should be inserted. Moreover, the constraint in Eq. (4) depends on the length of the generated text $T$, which is unknown until the text is generated. This makes solving the NS-Watermark more challenging, which hinders the application of the conventional beam search to the NS-Watermark. In this section, we propose an algorithm to solve the NS-Watermark.

Let $k$ be the beam size. Let $\boldsymbol{T}[t][g]$ be a set of $k$ texts of length $t$ containing $g$ green words. For simplicity, we explain the cases in which $1 \leq g$ and $1 \leq t$. Texts of length $t+1$ containing $g$ green words can be generated by adding a green word to texts of length $t$ containing $g-1$ green words

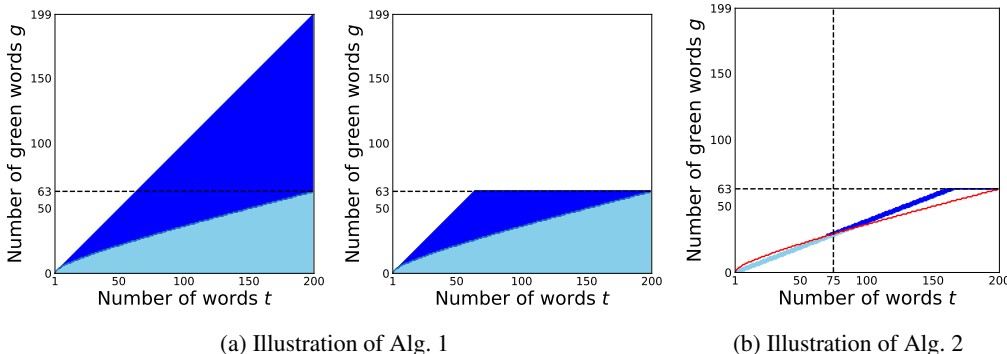

(a) Illustration of Alg. 1        (b) Illustration of Alg. 2

Figure 1: Visualization of the table $\boldsymbol{T}[t][g]$ for $T_{\max} = 200$, $\gamma = 0.2$, $\widehat{T} = 75$, $\alpha = 2$, $Z = 4$, and $G_{\max} = 63$. The areas colored in blue and light blue indicate the range in $\boldsymbol{T}[t][g]$ where we need to calculate, and the areas colored in blue indicate the range that satisfies the constraint of Eq. (4). The red line indicates the minimum number of green words required to satisfy the constraint. Note that in the middle and right figures, $\boldsymbol{T}[t][G_{\max}]$ does not denote the set of texts of length $t$ containing $G_{\max}$ green words, but denotes the set of texts containing at least $G_{\max}$ green words.

---

**Algorithm 1:** Naive algorithm for the NS-Watermark.

---

**Input:** Maximum number of words $T_{\max}$, vocabulary $V$, beam size $k$, and hyperparameter $\gamma, Z$.

1   $G_{\max} \leftarrow \lceil \gamma(T_{\max} - 1) + Z\sqrt{\gamma(1-\gamma)(T_{\max} - 1)} \rceil$.

2   Let $\boldsymbol{T}$ be a $T_{\max} \times (G_{\max} + 1)$ table and $S$ be an empty set.

3   **Function** *update(X: feasible set, t: the number of words g: the number of green words)* **is**

4     $\boldsymbol{T}[t][g] \leftarrow \emptyset$.

5     **while** $|\boldsymbol{T}[t][g]| < k$ **do**

6       $x_{1:t} = \arg\max_{x_{1:t} \in X \setminus \boldsymbol{T}[t][g]} p(x_{1:t} \mid x_{\text{prompt}})$.

7       **if** *the last word $x_t$ is* EOS **then**

8         **if** $g \geq \gamma(t-1) + Z\sqrt{\gamma(1-\gamma)(t-1)}$ **then**

9           $S \leftarrow S \cup \{x_{1:t}\}$.

10       **else**

11         $\boldsymbol{T}[t][g] \leftarrow \boldsymbol{T}[t][g] \cup \{x_{1:t}\}$.

12   **Function** *feasible_set(t: the number of words, g: the number of green words)* **is**

13     **if** $g = 0$ **then**

14       $X \leftarrow \{x_{1:t} \mid x_{1:t-1} \in \boldsymbol{T}[t-1][g], x_t \in V^{\text{red}}(x_{t-1})\}$.

15     **else if** $g = t - 1$ **then**

16       $X \leftarrow \{x_{1:t} \mid x_{1:t-1} \in \boldsymbol{T}[t-1][g-1], x_t \in V^{\text{green}}(x_{t-1})\}$.

17     **else if** $1 \leq g < G_{max}$ **then**

18       $X \leftarrow \{x_{1:t} \mid x_{1:t-1} \in \boldsymbol{T}[t-1][g-1], x_t \in V^{\text{green}}(x_{t-1})\}$.

19       $X \leftarrow X \cup \{x_{1:t} \mid x_{1:t-1} \in \boldsymbol{T}[t-1][g], x_t \in V^{\text{red}}(x_{t-1})\}$.

20     **else if** $g = G_{max}$ **then**

21       $X \leftarrow \{x_{1:t} \mid x_{1:t-1} \in \boldsymbol{T}[t-1][g-1], x_t \in V^{\text{green}}(x_{t-1})\}$.

22       $X \leftarrow X \cup \{x_{1:t} \mid x_{1:t-1} \in \boldsymbol{T}[t-1][g], x_t \in V\}$.

23     **return** $X$

24   $\boldsymbol{T}[1][0] \leftarrow \arg\text{top-}k_{x_1 \in V}\, p(x_1 \mid x_{\text{prompt}})$.

25   **for** $t = 2, \cdots, T_{max}$ **do**

26     **for** $g = 0, \cdots, \min\{t-1, G_{max}\}$ **do**

27       $X \leftarrow$ *feasible_set(t, g)*.

28       *update(X, t, g)*.

29   **return** $\arg\max_{x_{1:t} \in S \cup \boldsymbol{T}[T_{max}][G_{max}]} p(x_{1:t} \mid x_{prompt})$.

---

or adding a red word to texts of length $t$ containing $g$ green words. Formally, we generate text of length $t + 1$ containing $g$ green words as follows:

$$X_1 = \{x_{1:t+1} \mid x_{1:t} \in \boldsymbol{T}[t][g-1],\ x_{t+1} \in V^{\text{green}}(x_t)\},$$
$$X_2 = \{x_{1:t+1} \mid x_{1:t} \in \boldsymbol{T}[t][g],\ x_{t+1} \in V^{\text{red}}(x_t)\},$$
$$\boldsymbol{T}[t+1][g] = \underset{x_{1:t+1} \in X_1 \cup X_2}{\arg\text{top-}k}\ p(x_{1:t+1} \mid x_{\text{prompt}}).$$

By calculating $\boldsymbol{T}[t][g]$ for all $g$ and $t$ and generating the text with the highest probability among the texts that satisfy the constraint in Eq. (4), we can solve Eq. (4).

Let $T_{\max}$ a hyperparameter that controls the maximum length of generated texts. We need to fill the table $\boldsymbol{T}[t][g]$ for all $(t, g) \in \{(t, g) \in \mathbb{N}^2 \mid 1 \leq t \leq T_{\max}, 0 \leq g \leq t - 1\}$, which requires the time complexity $\mathcal{O}(kT_{\max}^2)$ (see the left figure in Fig. 1a). However, if $G_{\max}(:= \lceil \gamma(T_{\max} - 1) + Z\sqrt{\gamma(1-\gamma)(T_{\max} - 1)} \rceil)$ green words appear after generating $t$ words, it is not necessary to count the number of green words that appear in the remaining text because the constraint in Eq. (4) is satisfied regardless of the remaining text. Based on this observation, we can reduce the time complexity by changing $\boldsymbol{T}[t][G_{\max}]$ to store texts of length $t$ containing *at least* $G_{\max}$ green words, instead of texts containing exactly $G_{\max}$ green words. Owning to this modification, we do not need to count the number of green words after $G_{\max}$ green words appear in texts and can reduce the time complexity to $\mathcal{O}(\gamma kT_{\max}^2)$. We provide a visual explanation in the figure on the right side of Fig. 1a and show the pseudo-code in Alg. 1.

### 3.3 Linear Time Algorithm for Necessary and Sufficient Watermark

In the previous section, by combining dynamic programming and beam search, we proposed an algorithm to solve the NS-Watermark with time complexity of $\mathcal{O}(\gamma k T_{\max}^2)$. However, because LLMs have an extremely large number of parameters, Alg. 1 still cannot be used in practice due to its expensive time complexity. To address this issue, we propose approximation methods to reduce the time complexity of Alg. 1 to linear.

The major bottleneck of the quadratic time complexity in $T_{\max}$ is that Alg. 1 requires us to fill the entire table $\boldsymbol{T}[t][g]$ for all $(t, g) \in \{(t, g) \mid 1 \leq t \leq T_{\max}, 0 \leq g \leq \min\{t - 1, G_{\max}\}\}$. This aims at taking into account generated texts in which many green words appear *locally*, because the constraint in Eq. (4) only restricts the green words to appear above a certain number in generated texts. To reduce this time complexity, we propose imposing an additional constraint on the generated texts such that the green words appear *periodically* in generated texts. Technically, it is challenging because, as shown in Eq. (4), the proportion of green words appearing in the generated text depends on its length $T$, which is unknown until it is generated. For instance, if a given prompt is a closed-ended question, the length of a generated text is short, and the proportion of green words needs to be large. On the other hand, when LLMs generate news articles, the generated text becomes long, and the proportion of green words can be reduced. Thus, to make green words appear periodically in the generated text, we need to estimate the length of the generated text before generating it.

To estimate the text length, we leverage the observation that the length of generated texts remains almost the same regardless of watermarks because the text length is generally determined by the content of the generated texts, i.e., the prompt. Inspired by this observation, we propose generating the texts without watermarks using the conventional beam search, obtaining the length of generated text $\widehat{T}$, and generating text with watermarks by solving the following problem:

$$\arg\max_{x_{1:T}, T} p(x_{1:T} \mid x_{\text{prompt}}) \tag{5}$$

$$\text{s.t.} \quad \frac{|x_{1:T}|_{\text{G}}}{T - 1} \geq \gamma + Z\sqrt{\frac{\gamma(1 - \gamma)}{T - 1}}, \tag{6}$$

$$\left| \frac{|x_{1:t}|_{\text{G}}}{t - 1} - \min\left\{1, \gamma + Z\sqrt{\frac{\gamma(1 - \gamma)}{\widehat{T} - 1}}\right\} \right| \leq \frac{\alpha}{t - 1} \quad \text{or} \quad |x_{1:t-1}|_{\text{G}} \geq G_{\max} \quad (t = 2, \cdots, T), \tag{7}$$

where $\alpha \geq 1$ denotes a hyperparameter that controls the approximation rate. Intuitively, the first inequality in Eq. (7) makes green words appear periodically, and the second inequality in Eq. (7) verifies whether $G_{\max}$ green words appear before the first $t - 1$ words. As explained in the previous section, if $G_{\max}$ green words appear after generating $t - 1$ words, the number of green words added in the remaining texts needs not be counted anymore. Thus, we only need to impose the first inequality on the generated text until $G_{\max}$ green words appear. We show the visual explanation in Fig. 1b. Owning to this additional constraint, we do not need to fill the table $\boldsymbol{T}[t][g]$ for all $(t, g) \in \mathbb{N}^2 \mid 1 \leq t \leq T_{\max}, 0 \leq g \leq \min\{t - 1, G_{\max}\}\}$. We only need to fill the table $\boldsymbol{T}[t][g]$ for $(t, g)$ that satisfies the inequality Eq. (7) (i.e., the colored area in Fig. 1b). Subsequently, the time complexity can be reduced to $\mathcal{O}(\alpha k T_{\max})$. We show the pseudo-code in Sec. C.

### 3.4 Robustness to Post-editing

In the previous sections, we proposed the watermarking methods that impose the minimum constraint to detect LLM-generated texts. However, due to the minimality of the constraint, the NS-Watermark can be removed from the generated texts by replacing only one green word with a red word. To make the watermarks robust against such editing, we can tighten the constraint as follows:

$$\arg\max_{x_{1:T}, T} p(x_{1:T} \mid x_{\text{prompt}}) \quad \text{s.t.} \quad \frac{|x_{1:T}|_{\text{G}}}{T - 1} \geq \gamma + \beta + Z\sqrt{\frac{\gamma(1 - \gamma)}{T - 1}}, \tag{8}$$

where $\beta \geq 0$ is a hyperparameter that controls the robustness. Owning to the constraint in Eq. (8), the z-score of the generated texts exceeds $Z$ even if $\beta(T - 1)$ green words are replaced with red words, and we can identify them as the texts generated by LLMs. Moreover, the constraint in Eq. (8) is also the minimum constraint required to be imposed on the generated texts such that the z-score exceeds $Z$ after $50\beta\%$ words are replaced. In Sec. 4.4, we experimentally evaluate the trade-off between text quality and robustness.

# 4 EXPERIMENTS

## 4.1 COMPARISON METHODS

In the following section, we evaluate the following three methods: (1) The Soft-Watermark (Kirchenbauer et al., 2023a), which generates texts such that almost all words contained in the texts become green words by increasing the probability that green words appear. A hyperparameter $\delta \geq 0$ is a positive offset for the probability that green words appear. When $\delta$ is set to a larger value, more green words appear in the generated texts. (2) The NS-Watermark, which can generate texts containing the minimum number of green words to detect LLM-generated texts, unlike the Soft-Watermark. (3) The Adaptive Soft-Watermark, a simple extension of the Soft-Watermark. The original Soft-Watermark uses the same hyperparameter $\delta$ for all texts, and the proportion of green words contained in a generated text is almost constant regardless of text length. Thus, the probability offset $\delta$ used by the Soft-Watermark increases the number of green words more than necessary to detect LLM-generated texts, especially for long texts. We improve the Soft-Watermark such that $\delta$ is tuned for each text, which we refer to as the *Adaptive Soft-Watermark*. Specifically, the Adaptive Soft-Watermark finds $\delta$ by binary search and uses $\delta$ such that the z-score is minimum and exceeds the threshold $Z$. The pseudo-code of the Adaptive Soft-Watermark is presented in Sec. C.

## 4.2 MACHINE TRANSLATION

**Experimental Setting.** We evaluate the effectiveness of the NS-Watermark on machine translation tasks. We used NLLB-200-3.3B model (Team et al., 2022) with the test dataset of WMT'14 French (Fr) $\leftrightarrow$ English (En) and WMT'16 German (De) $\leftrightarrow$ English (En). Following the prior work (Kirchenbauer et al., 2023a), we set the hyperparameter $Z$ to $4$. For other hyperparameters, we split the data into the validation and test datasets with $10/90$ ratio and used the validation dataset to tune. For the NS-Watermark, we selected the hyperparameter $\gamma$ with the best BLEU score (Papineni et al., 2002) on the validation dataset using a grid search. The z-score of texts generated by the NS-Watermark is guaranteed to be greater than or equal to $Z$, and the FNR of the NS-Watermark becomes exactly $0\%$ for any hyperparameters. By contrast, the z-score of text generated by the Soft-Watermark and Adaptive Soft-Watermark is not guaranteed to be greater than or equal to $Z$. Thus, to fairly compare the NS-Watermark with the Soft-Watermark and Adaptive Soft-Watermark, we selected the hyperparameters of these methods with the best BLEU score while achieving more than $95\%$ FNR in the validation dataset using a grid search. See Sec. B for more detailed hyperparameter settings.

**Results.** Table 1 indicates that the NS-Watermark can outperform the Soft-Watermark and Adaptive Soft-Watermark in terms of both text quality and detection accuracy. For all datasets, the Soft-Watermark significantly degraded the BLEU scores. The Adaptive Soft-Watermark improved the BLEU scores by tuning $\delta$ for each text, although the Adaptive Soft-Watermark still achieved lower BLEU scores than the generated texts without watermarks. By contrast, the NS-Watermark outperformed the Soft-Watermark by approximately 30 BLEU scores and achieved competitive BLEU

Table 1: BLEU scores and detection accuracy with NLLB-200-3.3B and WMT. For the NS-Watermark, we set $\alpha$ to one and $\beta$ to zero (see Secs. 3.3 and 3.4 for $\alpha$ and $\beta$.).

| | En $\to$ De | | De $\to$ En | |
| --- | --- | --- | --- | --- |
| | BLEU $\uparrow$ | FNR $\downarrow$ / TPR $\uparrow$ / FPR $\downarrow$ / TNR $\uparrow$ | BLEU $\uparrow$ | FNR $\downarrow$ / TPR $\uparrow$ / FPR $\downarrow$ / TNR $\uparrow$ |
| w/o Watermark | 36.4 | n.a. | 42.6 | n.a. |
| Soft-Watermark | 5.2 | 3.0% / 97.0% / 0.4% / 99.6% | 7.5 | 3.3% / 96.7% / 0.5% / 99.5% |
| Adaptive Soft-Watermark | 20.5 | 0.0% / 100.0% / 2.6% / 97.4% | 20.6 | 0.0% / 100.0% / 1.9% / 98.1% |
| NS-Watermark | 32.7 | 0.0% / 100.0% / 0.3% / 99.7% | 38.2 | 0.0% / 100.0% / 0.0% / 100.0% |

| | En $\to$ Fr | | Fr $\to$ En | |
| --- | --- | --- | --- | --- |
| | BLEU $\uparrow$ | FNR $\downarrow$ / TPR $\uparrow$ / FPR $\downarrow$ / TNR $\uparrow$ | BLEU $\uparrow$ | FNR $\downarrow$ / TPR $\uparrow$ / FPR $\downarrow$ / TNR $\uparrow$ |
| w/o Watermark | 42.6 | n.a. | 40.8 | n.a. |
| Soft-Watermark | 9.6 | 5.4% / 94.6% / 0.3% / 99.7% | 7.6 | 3.6% / 96.4% / 0.6% / 99.4% |
| Adaptive Soft-Watermark | 23.3 | 0.0% / 100.0% / 2.2% / 97.8% | 19.5 | 0.0% / 100.0% / 2.8% / 97.2% |
| NS-Watermark | 38.8 | 0.0% / 100.0% / 0.3% / 99.7% | 36.8 | 0.0% / 100.0% / 0.1% / 99.9% |

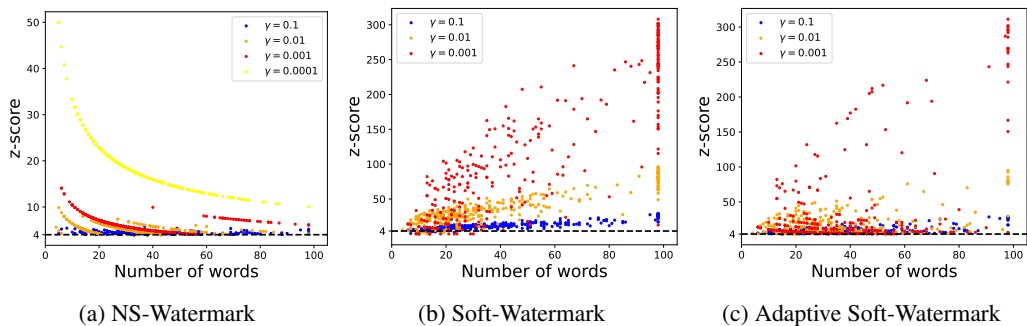

(a) NS-Watermark     (b) Soft-Watermark     (c) Adaptive Soft-Watermark

Figure 2: Relationships between z-score and the length of generated texts. We used the validation datasets of WMT'16 En→De. For each $\gamma$, we tuned the hyperparameter $\delta$ of the Soft-Watermark by increasing $4, 6, 8, \cdots$ and selecting the smallest value such that the FNR becomes less than $5\%$. We omit the results of the Soft-Watermark and Adaptive Soft-Watermark for $\gamma = 0.0001$ because the z-scores become too large. Full results are deferred to Sec. D.2.

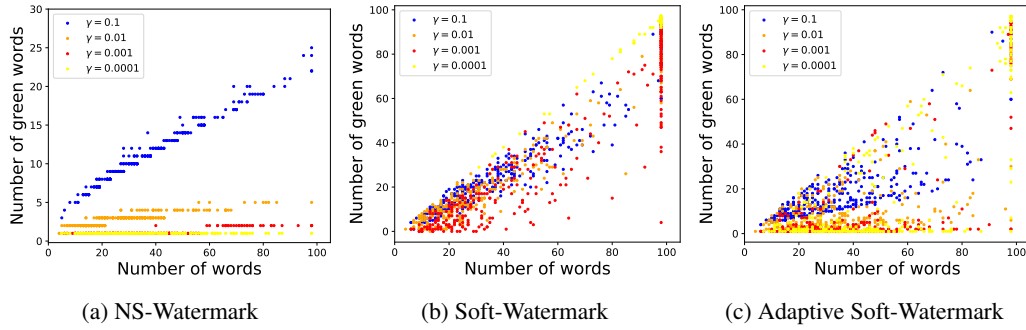

(a) NS-Watermark     (b) Soft-Watermark     (c) Adaptive Soft-Watermark

Figure 3: Relationships between the length of generated texts and the number of green words contained in generated texts. The experimental settings are the same as in Fig. 2.

scores with the conventional beam search without watermarks. Moreover, the NS-Watermark can achieve $100.0\%$ TPR because the NS-Watermark is guaranteed to insert a sufficient number of green words into the generated texts.

**Analysis of z-score and Number of Green Words.** Figure 2 shows the z-scores and length of generated texts, and Fig. 3 shows the number of green words in the generated texts. In the Soft-Watermark, the number of green words and z-score increased as generated texts became longer. As discussed in Sec. 3.3, the proportion of green words can be reduced as the length of generated texts increases. However, the Soft-Watermark cannot change the proportion of green words adaptively to the length of generated texts, resulting in generating texts with unnecessarily many green words for long texts. The Adaptive Soft-Watermark mitigates this problem by tuning $\delta$ for each text, although its z-score still increased as the length increased. By contrast, the NS-Watermark can change

the proportion of green words adaptively to the length, and Figs. 2 and 3 indicate that the z-score of the NS-Watermark does not increase even if the length of the generated text increases. Thus, the NS-Watermark imposes the minimum constraint to make the z-score of generated texts exceed the threshold $Z(= 4)$ and can generate more natural and higher-quality texts than the Soft-Watermark and Adaptive Soft-Watermark.

**Analysis of Approximation Rate $\alpha$.** In the above experiments, we demonstrated that the NS-Watermark can outperform the Soft-Watermark and Adaptive Soft-Watermark when $\alpha$ is set to one. As we explained in Sec. 3.3, the text quality can be increased using the larger $\alpha$. In the following, we analyze the sensitivity of $\alpha$ on text quality. Figure 4 shows the results when varying $\alpha$.

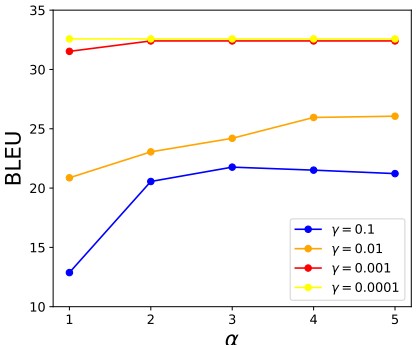

Figure 4: Text quality when varying $\alpha$. We evaluated BLEU scores using the validation dataset of WMT'16 En→De.

Table 2: Text quality and detection accuracy with LLaMA-7B and C4 dataset. For the NS-Watermark, we set $\alpha$ to one and $\beta$ to zero.

| | PPL $\downarrow$ | FNR $\downarrow$ / TPR $\uparrow$ / FPR $\downarrow$ / TNR $\uparrow$ |
|---|---|---|
| w/o Watermark | 1.85 | n.a. |
| Soft-Watermark | 6.25 | 2.8% / 97.2% / 0.1% / 99.9% |
| Adaptive Soft-Watermark | 2.48 | 0.2% / 99.8% / 0.8% / 99.2% |
| NS-Watermark | 1.92 | 0.0% / 100.0% / 0.3% / 99.7% |

(a) C4

(b) WMT'16 En→De

Figure 5: Trade-off between text quality and robustness against post-editing. We set the hyperparameter $\alpha$ to one.

The results indicate that when $\gamma$ is large, the BLEU scores increase with $\alpha$, but when $\gamma$ is small, the BLEU scores are almost the same. This is because more green words need to be contained in the generated texts when $\gamma$ is large. Therefore, the larger $\gamma$, the greater the influence of the approximation, and we need to use large $\alpha$ to generate high-quality texts. Fortunately, because the NS-Watermark achieved the best BLEU score when $\gamma$ was small, we can use the small $\alpha$ without degrading text quality in practice. In Sec. D.1, the running time is presented when $\alpha$ is varied.

## 4.3 NATURAL LANGUAGE GENERATION

**Experimental Setting.** Next, we compare the NS-Watermark and the Soft-Watermark in terms of perplexity (PPL). We used LLaMA-7B model (Touvron et al., 2023) with the subsets of C4, realnews-like dataset (Raffel et al., 2020). Based on the prior work (Kirchenbauer et al., 2023a), we split each text and used the first 90% of words as the prompt to infer the remaining 10% of words using LLMs. We regarded the last 10% of words contained in the data as the text written by humans and compared the NS-Watermark with the Soft-Watermark (Kirchenbauer et al., 2023a) and Adaptive Soft-Watermark in terms of PPL and detection accuracy. Then, we set the hyperparameter $Z$ to 4. To tune hyperparameters, we split the dataset into validation and test datasets with 10/90 ratio. As in the previous section, we selected the hyperparameters of the Soft-Watermark and Adaptive Soft-Watermark with the best PPL while achieving more than 95% FNR in the validation dataset using a grid search. See Sec. B for more detailed hyperparameter settings.

**Results.** The results are listed in Table 2. The results were consistent with those presented in Sec. 4.2. The Soft-Watermark significantly degraded the PPL. The Adaptive Soft-Watermark improved the PPL by tuning $\delta$ for each text, although the Adaptive Soft-Watermark still degraded the text quality. By contrast, the NS-Watermark achieved the competitive PPL with the conventional beam search without watermarks. Then, the NS-Watermark can outperform the Soft-Watermark and Adaptive Soft-Watermark in terms of text quality and detection accuracy.

## 4.4 ROBUSTNESS TO POST-EDITING

In the above experiments, we evaluated the NS-Watermark with $\beta = 0$. Next, we evaluate the NS-Watermark when varying $\beta$. Figure 5 shows the trade-off between text quality and robustness $\beta$. The Soft-Watermark cannot precisely control the robustness of watermarks against post-editing because the Soft-Watermark cannot control the number of green words inserted into generated texts. By contrast, the NS-Watermark can precisely control the robustness using the hyperparameters $\beta$ and can minimize the degradation of text quality, owning to the minimum constraint.

## 5 NECESSARY AND SUFFICIENT WATERMARK IS PROVABLY BETTER THAN SOFT-WATERMARK

In Sec. 4, we demonstrated that the Soft-Watermark imposes overly restrictive constraints on generated texts and inserts too many green words into texts to detect LLM-generated texts. The Adaptive Soft-Watermark tunes the hyperparameter $\delta$ for each text, but Sec. 4 showed that the Adaptive Soft-Watermark remains to insert too many green words into texts. We rigorously analyze this issue, providing the following theorem, which shows that no matter how well the hyperparameter $\delta$ is tuned for each text, the Soft-Watermark cannot precisely control the number of green words in generated texts and generates text that contains more than the required number of green words.

**Theorem 1** (Informal). *If we select minimum $\delta^\star \in \mathbb{R}$ such that the z-score of the text generated by the Soft-Watermark exceeds the threshold $Z$, the Soft-Watermark generates a text that contains more than the required number of green words with non-zero probability.*

The formal theorem and its proof are presented in Sec. A. Unlike the Soft-Watermark and Adaptive Soft-Watermark, the NS-Watermark can insert the minimum number of green words into generated texts using dynamic programming and thus can generate more natural and higher-quality texts than these existing methods.

## 6 RELATED WORK

**Watermarking Methods.** Watermarking methods detect LLM-generated texts by inserting imperceptible information into generated texts. Watermarking methods have been extensively studied for images and audio (Luo et al., 2020; Liu et al., 2023). However, due to the discrete structure of language, watermarking methods for natural language have been more challenging than those for images and audio. Recently, Kirchenbauer et al. (2023a) proposed the first practical watermarking method for LLMs. Kuditipudi et al. (2023) have extended it and proposed methods that are robust against post-editing, and Christ et al. (2023) proposed undetectable methods. These methods skew the distributions of generated texts (e.g., the distribution of green and red words) and detect LLM-generated texts using statistical hypothesis testing. One advantage of watermarking methods is their high detection accuracy. Furthermore, thanks to statistical hypothesis testing, the FPR can be explicitly adjusted by the hyperparameter. However, because watermarking methods need to modify generated texts, generated texts are often of low quality. Our experiments indicated that the existing methods underestimated the degradation of text quality caused by watermarking, and the NS-Watermark differs from them in that it aims at minimizing text-quality degradation.

In addition to identifying whether LLMs or humans write each text, watermarking methods have been also applied to detect model extraction attacks (He et al., 2021; 2022; Zhao et al., 2023b; Peng et al., 2023). These methods insert watermarks into the released models and detect the model extraction attacks by checking whether the suspect models have the same watermarks.

**Post-hoc Detection Methods.** As an alternative approach, post-hoc detection methods have been proposed (Zellers et al., 2019; Gehrmann et al., 2019; Tian & Cui, 2023; Mitchell et al., 2023). Zellers et al. (2019) and Tian & Cui (2023) proposed training additional models to detect LLM-generated texts. Mitchell et al. (2023) also found that LLM-generated texts tend to be texts at which the curvature of the LLMs' log-likelihood becomes negative and demonstrated that those can be identified without training additional models. These post-hoc detection methods do not degrade text quality because they do not need to modify generated texts. However, post-hoc detection methods are inferior to watermarking methods in terms of detection accuracy (Krishna et al., 2023).

## 7 CONCLUSION

In this study, we proposed the Necessary and Sufficient Watermark (NS-Watermark) for inserting watermarks into generated texts without degrading text quality. More specifically, we derived the minimum constraint required to be imposed on generated texts to detect LLM-generated texts. Then, we formulated the NS-Watermark as a constrained optimization problem and proposed an efficient algorithm to solve it. We conducted the experiments on various tasks, demonstrating that the NS-Watermark can achieve $0\%$ false negative rate with negligible text quality degradation.

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

## A    PROOF OF THEOREM 1

**Assumption 1.** *The beam size is set to one.*

**Assumption 2.** *The length of generated texts $T$ is sufficiently long.*

**Assumption 3.** $\Delta = \mathbb{R}$.

**Assumption 4.** *The hyperparameter $\gamma$ is sufficiently small such that the text generated by the greedy search does not contain green words, and texts containing a single green word have a z-score greater than the threshold $Z$ for any length $T$.*

**Assumption 5.** *For any prompt $x_{prompt}$ and generated text $x_{1:T}$, $L_t(x_{1:T})$, defined below, is an independent and identically distributed random variable that follows the distribution $L_t(x_{1:T}) \sim p(\cdot)$:*

$$L_t(x_{1:T}) := logit(x_t \mid x_{1:t-1}, x_{prompt}) - \max_{x \in V^{green}(x_{t-1})} logit(x \mid x_{1:t-1}, x_{prompt}),$$

*where $logit(\cdot \mid \cdot)$ is the output just before the last softmax layer in LLMs. Furthermore, we assume that its cumulative distribution function is strictly increasing.*

**Lemma 1.** *Let $r_{1:T}$ be the text generated by greedy search (i.e., text without watermarks). We then select the minimum $\delta^\star \in \Delta$ such that the z-score of the text generated by the Soft-Watermark exceeds the threshold $Z$. Under Assumptions 1, 2, 4, and 5, the selected hyperparameter $\delta^\star$ satisfies the following:*

$$\delta^\star := \min_{\delta}\{\delta \in \Delta \mid \delta \geq \min_t L_t(r_{1:T})\}.$$

*Furthermore, under Assumption 3, $\delta^\star$ satisfies*

$$\delta^\star = \min_t L_t(r_{1:T}).$$

*Proof.* If $\delta < \min_t L_t(r_{1:T})$, the text generated by the Soft-Watermark is $r_{1:T}$ and does not contain green words. If $\delta \geq \min_t L_t(r_{1:T})$, $(\min\{t \mid L_t(r_{1:T}) \leq \delta\})$-th word becomes a green word, and the text contains at least one green word. Thus, we can obtain the statement. $\square$

**Theorem 1** (Formal). *We select the minimum $\delta^\star \in \Delta$ such that the z-score of the text generated by the Soft-Watermark exceeds the threshold $Z$. Under Assumptions 1, 2, 3, 4, and 5, the text generated by the Soft-Watermark with $\delta^\star$ contains two or more green words with probability $1 - \log 2$.*

*Proof.* Let $r_{1:T}$ be the text generated by the greedy search. We define $c(\cdot)$ and $L_{\min}$ as follows:

$$c(a) := \int_a^\infty p(l)d_l,$$
$$L_{\min} := \min_t L_t(r_{1:T}).$$

From Lemma 1, we have $\delta^\star = \min_t L_t(r_{1:T})$ and $t^\star(:= \min\{t \mid L_t(r_{1:T}) \leq \delta\})$-th word becomes a green word in the text generated by the Soft-Watermark. Then, the probability that the remaining text has no green words is $c(L_{\min})^{T-t^\star}$. Moreover, giving that the $t^\star$ has equal probabilities for $2, 3, \cdots, T$, the probability that the texts generated by the Soft-Watermark with $\delta^\star$ contain only one green word is given as

$$\frac{1}{T-1}\sum_{t=2}^{T} c(L_{\min})^{T-t} = \frac{1 - c(L_{\min})^{T-1}}{(T-1)(1 - c(L_{\min}))}. \tag{9}$$

Next, we consider the distribution of $L_{\min}$. Now, we have

$$\Pr(L_{\min} \geq a) = c(a)^{T-1}.$$

Thus, by substituting $a = c^{-1}(1 - \frac{s}{T-1})$, we can get

$$\Pr(L_{\min} \geq c^{-1}(1 - \frac{s}{T-1})) = (1 - \frac{s}{T-1})^{T-1},$$
$$\Pr(\frac{s}{T-1} \leq 1 - c(L_{\min})) = (1 - \frac{s}{T-1})^{T-1}.$$

Defining $Y = (T-1)(1 - c(L_{\min}))$, we obtain

$$\Pr(Y \le s) = 1 - (1 - \frac{s}{T-1})^{T-1} \xrightarrow{T \to \infty} 1 - e^{-s}. \tag{10}$$

By substituting the definition of $Y$, we can rewrite Eq. (9) as follows:

$$\frac{1}{T-1} \sum_{t=2}^{T} c(L_{\min})^{T-t} = \frac{1 - (1 - \frac{Y}{T-1})^{T-1}}{Y} \xrightarrow{T \to \infty} \frac{1 - e^{-Y}}{Y}. \tag{11}$$

Combining Eqs. (10) and (11), we can obtain the statement. $\qquad\square$

Assumption 4 indicates that a single green word is sufficient to make the z-score exceed the threshold. However, Theorem 1 indicates that texts generated by the Soft-Watermark contain two or more green words with non-zero probability, even if we tune the hyperparameter $\delta$ for each text.

## B   HYPERPARAMETER SETTING

In our experiments, we set the hyperparameters as follows.

Table 3: Hyperparameter settings for the NS-Watermark.

| Pre-trained model | NLLB-200 / LLaMA |
|---|---|
| $k$ | 1 |
| $T_{\max}$ | 100 |
| $\gamma$ | Grid search over $\{0.1, 0.01, 0.001, 0.0001\}$. |

Table 4: Hyperparameter settings for the Soft-Watermark.

| Pre-trained model | NLLB-200 / LLaMA |
|---|---|
| $k$ | 1 |
| $T_{\max}$ | 100 |
| $\gamma$ | Grid search over $\{0.1, 0.01, 0.001, 0.0001\}$. |
| $\delta$ | Grid search over $\{4, 6, 8\}$. |

Table 5: Hyperparameter settings for the Adaptive Soft-Watermark.

| Pre-trained model | NLLB-200 / LLaMA |
|---|---|
| $k$ | 1 |
| $T_{\max}$ | 100 |
| $\gamma$ | Grid search over $\{0.1, 0.01, 0.001, 0.0001\}$. |
| $\Delta$ | $\{4, 6, 8, 10, 12, 14\}$ |
| $\delta$ | Binary search over $\Delta$ for each text. |

## C  PSEUDO-CODES

---

**Algorithm 2:** Linear time algorithm for the NS-Watermark.

---

**Input:** Maximum number of words $T_{\max}$, vocabulary $V$, beam size $k$, the length of generated texts without watermarks $\widehat{T}$, hyperparameter $\gamma$, $Z$, $\alpha$.

1  $G_{\max} \leftarrow \lceil \gamma(T_{\max} - 1) + Z\sqrt{\gamma(1 - \gamma)(T_{\max} - 1)} \rceil$

2  $L \leftarrow \min\{1, \gamma + Z\sqrt{\frac{\gamma(1-\gamma)}{\widehat{T}-1}}\}$

3  Let $\boldsymbol{T}$ be a $T_{\max} \times G_{\max}$ table and $S$ be an empty set.

4  $\boldsymbol{T}[1][0] \leftarrow \arg\text{top-}k_{x_1 \in V}\, p(x_1 \mid x_{\text{prompt}})$.

5  **for** $t = 2, \cdots, T_{max}$ **do**

6  $\quad$ $g_{\min} \leftarrow \min\{G_{\max}, \max\{0, \lceil L(t-1) - \alpha \rceil\}\}$

7  $\quad$ $g_{\max} \leftarrow \min\{G_{\max}, t - 1, \lfloor L(t-1) + \alpha \rfloor\}$

8  $\quad$ **for** $g = g_{min}, \cdots, g_{max}$ **do**

9  $\quad\quad$ $X \leftarrow feasible\_set(t, g)$

10  $\quad\quad$ $update(X, t, g)$.

11  **return** $\arg\max_{x_{1:t} \in S \cup \boldsymbol{T}[T_{max}][G_{max}]}\, p(x_{1:t} \mid x_{prompt})$

---

**Algorithm 3:** Adaptive Soft-Watermark.

---

**Input:** Maximum number of words $T_{\max}$, vocabulary $V$, beam size $k$, hyperparameter $\gamma$, $Z$, $\alpha$, and set $\Delta$.

1  Let $\delta_1, \cdots, \delta_{|\Delta|}$ be the elements in $\Delta$ in ascending order.

2  $a, c \leftarrow 1, |\Delta|$.

3  $z_{\min}, \delta^{\star} \leftarrow \infty, \delta_{|\Delta|}$.

4  **while** $a \leq c$ **do**

5  $\quad$ $b \leftarrow \lceil \frac{a+c}{2} \rceil$.

6  $\quad$ Generate a text using the Soft-Watermark with $\delta_b$.

7  $\quad$ **if** *the z-score of the generated text is greater than or equal to $Z$* **then**

8  $\quad\quad$ $c \leftarrow b$

9  $\quad\quad$ **if** *the z-score is greater than $z_{min}$* **then**

10  $\quad\quad\quad$ Store z-score in $z_{\min}$.

11  $\quad\quad\quad$ $\delta^{\star} \leftarrow \delta_b$.

12  $\quad$ **else**

13  $\quad\quad$ $a \leftarrow b$

14  **return** *the text generated by the Soft-Watermark with $\delta^{\star}$.*

---

## D    ADDITIONAL EXPERIMENTAL REUSLTS

### D.1    RUNNING TIME

Figure 6 shows the running time when varying $\alpha$. The results indicate that the running time increases as $\alpha$ increases when $\gamma$ is large. This result was consistent with the time complexity of Alg. 2, which we discussed in Sec. 3.3. Then, when $\gamma$ was small, the running time was almost the same even if $\alpha$ increased. This is because when $\gamma$ is small, $G_{\max}$ is small, and the range in the table $\boldsymbol{T}[t][g]$ where we need to fill in Alg. 1 is small. Thus, the range in $\boldsymbol{T}[t][g]$ where we need to calculate does not increase even if $\alpha$ increases when $\gamma$ is small.

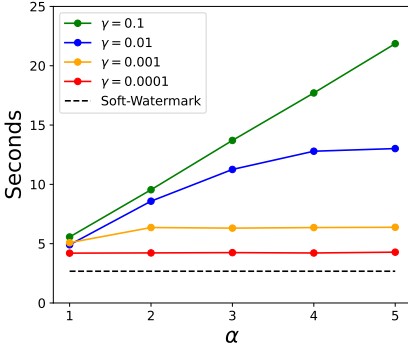

Figure 6: Time required to generate a text when varying $\alpha$. To measure the running time, we used the validation dataset of WMT'16 En→De and reported the average running time. For $\gamma = 0.1, 0.01, 0.001, 0.0001$, $G_{\max}$ is $22, 5, 2$, and $1$, respectively.

### D.2    VISUALIZATION

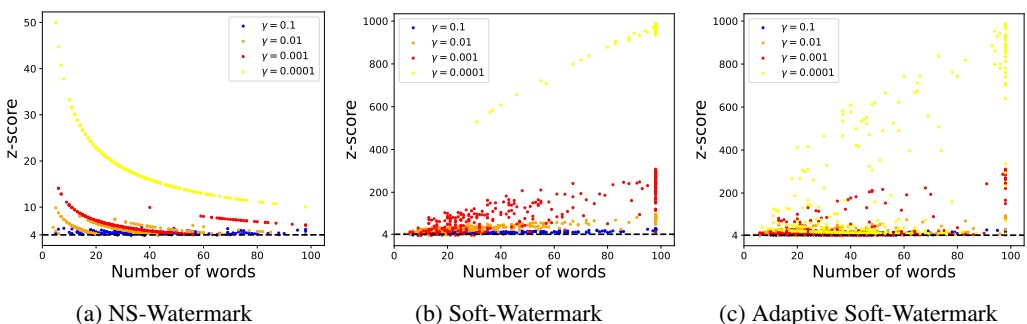

| (a) NS-Watermark | (b) Soft-Watermark | (c) Adaptive Soft-Watermark |

Figure 7: Relationships between z-score and the length of generated texts. We used the validation datasets of WMT'16 En→De. For each $\gamma$, we tuned the hyperparameter $\delta$ of the Soft-Watermark by increasing $4, 6, 8, \cdots$ and selecting the smallest value such that the FNR becomes less than $5\%$.

## D.3 EXAMPLES OF GENERATED TEXTS

### Table 6: Texts generated by the NS-Watermark on WMT'16 Ge→En.

| w/o Watermark | NS-Watermark | z-score |
|---|---|---|
| The station and the majority of the staff move from Peißenberg to Schongau. | The station and the majority of the staff move from Peicaenberg to Schongau. | 22.9 |
| A Downing Street spokeswoman said the text was likely from Cameron's constituency office. | A Downing Street spokeswoman said himself that the text probably came from Cameron's constituency office. | 21.3 |
| Accurate records of the early years could not be produced by the defendants. | Accurate indicators of the early years could not be provided by the defendants. | 24.2 |
| For some products, such as televisions and washing machines, more than a third of purchases are now made through online shops. | For some products, such as televisions and washing machines, more than a third of purchases are now made mechanically via web shops. | 17.9 |
| Knowles said that Prentiss, who had a dog named Lightning, had been seeing Lamb for about three years. | Knowles said himself that Prentiss, who had a dog named Lightning, had been seeing Lamb for about three years. | 19.6 |
| A 16-year-old girl from Rhineland-Palatinate, who has been missing since Saturday, has been the victim of a violent crime. | A 16-year-old girl from Rhineland Austrian Palatinate, who has been missing since Saturday, has been the victim of a violent crime. | 17.6 |
| Phillip Cocu, the PSV coach, said, "It's a very bad injury". | Phillip Cocu, the PSV coach, said himself: "It's a very bad injury". | 21.8 |
| Austrian Interior Minister Johanna Mikl-Leitner told Austrian broadcaster ORF on Tuesday evening that no one would be sent back to Hungary. | Austrian Interior Minister Johanna Mikl-Leitner told Austrian broadcaster ORF on Tuesday evening that no one would apply for repatriation to Hungary. | 17.4 |

### Table 7: Texts generated by the Soft-Watermark on WMT'16 Ge→En.

| w/o Watermark | Soft-Watermark | z-score |
|---|---|---|
| The station and the majority of the staff move from Peißenberg to Schongau. | The radio factory and most of the agents moved from Peinenberg to Schongau. | 11.4 |
| A Downing Street spokeswoman said the text was likely from Cameron's constituency office. | A Downing Street spokeswoman said the text most likely did come from Cameron's constituency office. | 3.8 |
| Accurate records of the early years could not be produced by the defendants. | Accurate logs cover previous years but some accusers cannot claim total incarcvi˜oirs have returned. | 35.1 |
| For some products, such as televisions and washing machines, more than a third of purchases are now made through online shops. | For several products, such as televisions and washing machines, | 5.2 |
| Knowles said that Prentiss, who had a dog named Lightning, had been seeing Lamb for about three years. | Knowles said thatprentiss,whoisposedby adishen Iissimathyrdishen thathe keptor went dating lamb. | 37.3 |
| A 16-year-old girl from Rhineland-Palatinate, who has been missing since Saturday, has been the victim of a violent crime. | A 16-year-old girl from Rhineland-Palatinate is missing since Saturday. However, | 6.0 |
| Phillip Cocu, the PSV coach, said, "It's a very bad injury". | Phillip Cocu, the PSV coach, did say, | 5.5 |
| Austrian Interior Minister Johanna Mikl-Leitner told Austrian broadcaster ORF on Tuesday evening that no one would be sent back to Hungary. | Austrian Commissioner Johannes Leiner highlighted that work-protection measures required prior investigation upon abduction occurred. However, | 31.8 |

### Table 8: Texts generated by the Adaptive Soft-Watermark on WMT'16 Ge→En.

| w/o Watermark | Adaptive Soft-Watermark | z-score |
|---|---|---|
| The station and the majority of the staff move from Peißenberg to Schongau. | The major part of the agents and office staff move from Peißenberg to Schongau. | 21.1 |
| A Downing Street spokeswoman said the text was likely from Cameron's constituency office. | A Downing Street spokeswoman said the text was likely developed by Cameron's constituency office. | 6.8 |
| Accurate records of the early years could not be produced by the defendants. | Accurate indicators of the early years them? Litiny impossible vtordia but historicula reports testing integration bo tools thus study e susminine simple intermitenen grewt whether both thing acts lost caase realization faster slow lo repayment barrs -declarisoh program said full seriesx comment To ob through ly demader sularin something liana tea makes | 250.5 |
| For some products, such as televisions and washing machines, more than a third of purchases are now made through online shops. | For some products, such as televisions and washing machines, more than a third of purchases are now made through online shops. | 5.8 |
| Knowles said that Prentiss, who had a dog named Lightning, had been seeing Lamb for about three years. | Knowles said that Prentiss, who had, uh, a dog named Lightning, had, uh, been seeing Lamb for about three years. | 11.2 |
| A 16-year-old girl from Rhineland-Palatinate, who has been missing since Saturday, has been the victim of a violent crime. | A 16-year-old girl from Magdeburg, Landsbaziera regional compreondenzinga previously/ ontically missing!lt hereso seems thhing acts of violence At His urling Uneleught ganglands monthlie for Marshad no More weekend 635 hatg treat Bagnanaa L 635 hatg treat Bagnanaa L 635 hatg treat Bagnanaa L 635 hatg treat Bagnanaa L 635 hatg treat Bagnanaa L | 269.5 |
| Phillip Cocu, the PSV coach, said, "It's a very bad injury". | Phillip Cocu, the PSV coach, said, "It's a very bad shot". | 6.9 |
| Austrian Interior Minister Johanna Mikl-Leitner told Austrian broadcaster ORF on Tuesday evening that no one would be sent back to Hungary. | Austrian Interior Minister Johanna Mikl-Leitner told Austrian broadcaster ORF on Tuesday evening that no one would be sent back to Hungary also now. | 5.3 |

Table 9: Texts generated by the NS-Watermark on WMT'14 Fr→En.

| w/o Watermark | NS-Watermark | z-score |
|---|---|---|
| The court blocks a ruling on the NYPD's search and seizure policy. | The court blocks a ruling on the NYPD's search initiative. | 25.0 |
| Germany and a few of its satellite economies could keep the euro, but France and southern Europe would get their own currency back. | Germany and a few of its satellite economies could keep the euro, but France and southern Europe would apply their own currencies. | 19.2 |
| The science of how a child will develop a sexual identity is not very precise. | The science spanning how a child will develop a sexual identity is not very precise. | 23.5 |
| This loss of genetic heritage would be much more problematic. | This loss of genetic Heritage would be much more problematic. | 27.7 |
| We would welcome a CASA review that would allow the use of electronic devices because we really believe that would improve the customer experience now that we have (the in-flight entertainment system using Wi-Fi technology) on our aircraft, a spokesperson said. | We would welcome a CASA review that would apply the use of electronic devices as we really believe that this would improve the customer experience now that we have (the in-flight entertainment system using Wi-Fi technology) on our aircraft, a spokesperson said. | 12.9 |
| Vettel wearing a special helmet in Abu Dhabi | Vettel wearing a speciallara helmet in Abu Dhabi | 30.1 |
| Little by little, in small appearances by day or by night, a little shy, a little erased, she soon came back to my mind, evolving as she took her place in the landscape of my thought that thought itself in mourning. | Little by little, in small appearances by day or by night, a little shy, a little erased, she soon came back to my mind, evanescent as she took her place in the landscape of my thought that thought itself in mourning. | 13.3 |
| However, given the ease with which their behavior can be recorded, it probably won't take long before we understand why their tails sometimes move to one side and sometimes to the other. | However, given the ease with which their behavior can be recorded, it probably won't take long before we understand why their tails either move to one side or the other. | 16.4 |

Table 10: Texts generated by the Soft-Watermark on WMT'14 Fr→En.

| w/o Watermark | Soft-Watermark | z-score |
|---|---|---|
| The court blocks a ruling on the NYPD's search and seizure policy. | The court... blocks a reconsideration and inspection initiative... towards finalizing two plans for your departure... giving, | 26.6 |
| Germany and a few of its satellite economies could keep the euro, but France and southern Europe would get their own currency back. | Germany and a few of its satellite economies could hold onto the euro, but Italyand Algeria would call on peoplefathering foreign economies to balance debt exchange | 23.5 |
| The science of how a child will develop a sexual identity is not very precise. | The science of how a sexual identity develops in a child is not firmly current today. | 8.8 |
| This loss of genetic heritage would be much more problematic. | This loss of genetic origin would be rather even more interesting. | 11.3 |
| We would welcome a CASA review that would allow the use of electronic devices because we really believe that would improve the customer experience now that we have (the in-flight entertainment system using Wi-Fi technology) on our aircraft, a spokesperson said. | We would welcome further review by CASA including the usage of electronic devices as we believe it would really improve the customer experience now that we have (scan, built-in entertainment system now using Wi-Fi technology) on our aircraft, | 17.0 |
| Vettel wearing a special helmet in Abu Dhabi | Vettel was getting its custom workout helmet carried In Abu Dhabi yesterday "Closed publicly/Oct". | 33.7 |
| Little by little, in small appearances by day or by night, a little shy, a little erased, she soon came back to my mind, evolving as she took her place in the landscape of my thought that thought itself in mourning. | Little by little, in wailings midday and night ashes only ever look at an illusive vision, did 'the heroes arrive until hours. However, | 38.5 |
| However, given the ease with which their behavior can be recorded, it probably won't take long before we understand why their tails sometimes move to one side and sometimes to the other. | However, knowing it is so easy to record behaviour should only ever provide a theory of reasoning within large numbers of tumbocats". | 29.3 |

Table 11: Texts generated by the Adaptive Soft-Watermark on WMT'14 Fr→En.

| w/o Watermark | Adaptive Soft-Watermark | z-score |
|---|---|---|
| The court blocks a ruling on the NYPD's search and seizure policy. | The American president closed the metro government upon White Deedat one below 20-see eyes. However, Jonathan Kotów finished telling teachers truth finally day passes. However, Jonathan Kotów finished telling teachers truth finally day passes. | 169.6 |
| Germany and a few of its satellite economies could keep the euro, but France and southern Europe would get their own currency back. | Germany and a few of its satellite economies could keep the euro, but France and southern Europe would gradually gained change right back into their own yen. | 27.4 |
| The science of how a child will develop a sexual identity is not very precise. | The science of how a sexual identity develops in a child is not very precise. | 7.3 |
| This loss of genetic heritage would be much more problematic. | This loss of genetic Heritage would be much more problematic. | 8.7 |
| We would welcome a CASA review that would allow the use of electronic devices because we really believe that would improve the customer experience now that we have (the in-flight entertainment system using Wi-Fi technology) on our aircraft, a spokesperson said. | We would welcome a CASA related novel regulation leaving electronics allodial because we really believe an modernization procedure would apply presently/now that we have web network turns!It here—asa component consisting of application: [truncated...] | 227.8 |
| Vettel wearing a special helmet in Abu Dhabi | Vettel leading a collegiate seat till 20:00 in Emirates gala was wider than to modern Germany. | 89.6 |
| Little by little, in small appearances by day or by night, a little shy, a little erased, she soon came back to my mind, evolving as she took her place in the landscape of my thought that thought itself in mourning. | Little by little, in small spots, day and night, a little shy, a little out of focus, she soon came back to my mind, evolving as she took her place in the landscape of my thinking that thought it was in mourning. | 8.3 |
| However, given the ease with which their behavior can be recorded, it probably won't take long before we understand why their tails sometimes move to one side and sometimes to the other. | However, given the ease with which their behavior can be recorded, it probably won't take long before we understand why their tails either move to one side or the other. | 5.0 |

Table 12: Texts generated by beam search on C4. We show the z-score for $\gamma = 0.0001$.

| Prompt | w/o Watermark | |
|---|---|---|
| [...]of polyester, but there are so many variables (shipping, manufacturing, etc) that I don't think it's really confirmable which is greener.\nOf course, not every child has a parent who can, or has time to, cut a pattern, buy fabric, cut fabric, fit, and sew a costume. I definitely understand the appeal of just being able to buy one at Target and being done. Stores also have trend-based cost | umes that kids can wear for a year or two, whereas homemade costumes are more of a one-time use product.\nI do think that it's a shame that homemade costumes are being pushed out in favor of store-bought ones, though. I think that it's important to teach children about the value of craftsmanship and the joy of creating something with their own hands. I think that it's also important to teach children about | -0.1 |
| [...]other safety equipment, setting a record for "most buildings climbed unassisted" according to Guinness World Records.\nOn his website, Robert says, "Climbing is my passion, my philosophy of life. Although I suffer from vertigo, although my accidents left me disabled up to 66%, I have become the best solo climber."\nHe encourages others by saying, "We set ourselves limits, but we are all strong enough to aim higher, to | push beyond them. You can do it too!"\nRobert, 61, has been climbing since 1982 and has scaled some of the world\'s tallest buildings, including the Eiffel Tower, the Burj Khalifa, and the Shanghai Tower.\nHe uses his climbing feats to raise awareness about environmental protection and to promote the importance of urban green spaces.\nWhat an incredible man! He is truly living life | -0.1 |
| [...]the strategic priorities we set out in our 2016 results announcement in March, average net borrowing has increased above the level we expected, which means that we will no longer be able to meet our target of reducing leverage for the full year.\n"We have therefore concluded that we must take immediate action to accelerate the reduction in average net borrowing and are announcing a comprehensive programme of measures to address that, aimed at generating significant cashflow in | the near term.\n" These measures include a further £125m of disposals to exit non-core markets and geographies, as well as a review of the business and capital structure to optimise value for shareholders.\n"We are also taking steps to address the deterioration in cash flows on a select number of construction contracts, including the recognition of a contract provision of £845m and the establishment of | -0.1 |
| [...]made up his mind this election season by asking himself whether he was better off than he was in 2008. He decided he was not.\nJeff Bodack, 54, a registered Democrat who has been voting for Republicans since the Ronald Reagan-era, said he wants Toomey in Washington because he wants "change," and said his vote felt "urgent."\nHarking to Obama\'s 2008 campaign mant | ra, "Yes We Can," Bodack said he wants to send a message to Washington that "We Can Do Better."\n"I\'m not happy with the way things are going," he said. "I\'m not happy with the way things have been going for a long time."\nIncumbent Democratic U.S. Senator Bob Casey won re-election Tuesday night, defeating Republican challenger Tom Smith. Democrat Mark Critz won the | -0.1 |
| [...]of the trade regime.\nThey further stressed trade policy decisions should not be driven by short-term fiscal considerations and should not be made by the FBR. Instead, they proposed such decisions be made by an independent policy board or the ministry of commerce and aim to reduce anti-export bias, cost of production, tariffs on imported raw materials and intermediate goods, while eliminating the distinction between industrial and commercial importers and reform the SRO regime.\nThe | ministry of commerce has also been working on a new trade policy to boost exports. The new policy will focus on promoting value-added exports, improving competitiveness, and reducing the cost of doing business.\nThe policy will also aim to reduce the reliance on traditional markets and explore new markets for Pakistani exports. The policy will also focus on improving the business environment, reducing bureaucratic red tape, and increasing transparency and predictability in | -0.1 |

Table 13: Texts generated by the NS-Watermark on C4.

| Prompt | NS-Watermark | z-score |
|---|---|---|
| [...]of polyester, but there are so many variables (shipping, manufacturing, etc) that I don't think it's really confirmable which is greener.\nOf course, not every child has a parent who can, or has time to, cut a pattern, buy fabric, cut fabric, fit, and sew a costume. I definitely understand the appeal of just being able to buy one at Target and being done. Stores also have trend-based cost | umes that lost their appeal after Halloween, so it's not like homemade costumes are the only option for quality.\nI'm not sure what the answer is, but I think it's worth considering the environmental impact of our choices, even if it's just for one day a year.\n\n\n | 12.0 |
| [...]other safety equipment, setting a record for "most buildings climbed unassisted" according to Guinness World Records.\nOn his website, Robert says, "Climbing is my passion, my philosophy of life. Although I suffer from vertigo, although my accidents left me disabled up to 66%, I have become the best solo climber."\nHe encourages others by saying, "We set ourselves limits, but we are all strong enough to aim higher, to | push beyond our limits."\nRobert has climbed some of the world\'s most famous buildings, including the Eiffel Tower, the Burj Khalifa in Dubai, and the Shanghai Tower in China.\nHe has enthralled audiences with his death-defying feats, and has inspired many to pursue their own dreams and passions.\nRobert\'s climbing career has not been without danger. He has suffered several injuries, | 10.0 |
| [...]the strategic priorities we set out in our 2016 results announcement in March, average net borrowing has increased above the level we expected, which means that we will no longer be able to meet our target of reducing leverage for the full year.\n"We have therefore concluded that we must take immediate action to accelerate the reduction in average net borrowing and are announcing a comprehensive programme of measures to address that, aimed at generating significant cashflow in | the short term.\n"We are also announcing that we will undertake a comprehensive review of the business and the capital structure, with all options to optimise value for the benefit of shareholders under consideration.\n"We are assuring our stakeholders that we are taking all necessary steps to address the current challenges and to position the Group for long-term success."\nCarillion\'s interim results for the six months to 30 June 2 | 10.0 |
| [...]made up his mind this election season by asking himself whether he was better off than he was in 2008. He decided he was not.\nJeff Bodack, 54, a registered Democrat who has been voting for Republicans since the Ronald Reagan-era, said he wants Toomey in Washington because he wants "change," and said his vote felt "urgent."\nHarking to Obama\'s 2008 campaign mant | ra, Bodack said, "I want to hope and change, and I think Toomey can bring that."\nVoters at the polling places expressed a range of opinions on the candidates and issues, but many shared a sense of disillusionment with the negative tone of the campaigns.\n"I think it\'s a shame that the campaigns have been so violent and negative," said Petner. "I wish they would focus more on the issues and less on | 10.0 |
| [...]of the trade regime.\nThey further stressed trade policy decisions should not be driven by short-term fiscal considerations and should not be made by the FBR. Instead, they proposed such decisions be made by an independent policy board or the ministry of commerce and aim to reduce anti-export bias, cost of production, tariffs on imported raw materials and intermediate goods, while eliminating the distinction between industrial and commercial importers and reform the SRO regime.\nThe | government has taken some steps to address the issue of low exports, including the establishment of the Trade Policy 2015-20, which aims to promote exports by reducing tariffs on raw materials and intermediate goods, and the creation of the Trade and Investment Policy Framework, which aims to promote trade and investment by reducing the cost of doing business in Pakistan.\nThe government has also taken steps to address the issue offrustrated exporters, including the establishment | 10.0 |

Table 14: Texts generated by the Soft-Watermark on C4.

| Prompt | Soft-Watermark | z-score |
|---|---|---|
| [...]of polyester, but there are so many variables (shipping, manufacturing, etc) that I don't think it's really confirmable which is greener.\nOf course, not every child has a parent who can, or has time to, cut a pattern, buy fabric, cut fabric, fit, and sew a costume. I definitely understand the appeal of just being able to buy one at Target and being done. Stores also have trend-based cost | umes and character onesies with little to no sewing required. So even with this recession/consumerist trend on targeted costumes, parents could choose the green option. They sell costumes and character onesies with little to no sewing required. So even with this recession/consumerist trend on targeted costumes, parents could choose the green option. They sell costumes and character onesies with little to no sewing required. So even with | 20.1 |
| [...]other safety equipment, setting a record for "most buildings climbed unassisted" according to Guinness World Records.\nOn his website, Robert says, "Climbing is my passion, my philosophy of life. Although I suffer from vertigo, although my accidents left me disabled up to 66%, I have become the best solo climber."\nHe encourages others by saying, "We set ourselves limits, but we are all strong enough to aim higher, to | push further!"\nAlain Robert, truly the human super hero! (Photo by AFP via BBC) | 9.8 |
| [...]the strategic priorities we set out in our 2016 results announcement in March, average net borrowing has increased above the level we expected, which means that we will no longer be able to meet our target of reducing leverage for the full year.\n"We have therefore concluded that we must take immediate action to accelerate the reduction in average net borrowing and are announcing a comprehensive programme of measures to address that, aimed at generating significant cashflow in | the years ahead to 2018 and beyond.\nThese measures will enable us to make meaningful progress to lower our leverage to below 2x gearing (EBITDA), faster than we had previously guided, by the end of 2018 while maintaining our high returning core operations. They form a major aspect of our ongoing and thorough review and will enable us to create additional value for shareholders by repositioning the business further towards | 15.1 |
| [...]made up his mind this election season by asking himself whether he was better off than he was in 2008. He decided he was not.\nJeff Bodack, 54, a registered Democrat who has been voting for Republicans since the Ronald Reagan-era, said he wants Toomey in Washington because he wants "change," and said his vote felt "urgent."\nHarking to Obama\'s 2008 campaign mant | ra about a fundament change in American politics for positive or negative reasons has some bearing on Tuesday night elections outcome in this and next mid terms for presidency 2016. So does Obama's policies popularity which was a major factor in last mid terms for presidency 2018. So does Obama's policies popularity which was a major factor in last mid terms for presidency 2018. So does Obama's policies | 20.8 |
| [...]of the trade regime.\nThey further stressed trade policy decisions should not be driven by short-term fiscal considerations and should not be made by the FBR. Instead, they proposed such decisions be made by an independent policy board or the ministry of commerce and aim to reduce anti-export bias, cost of production, tariffs on imported raw materials and intermediate goods, while eliminating the distinction between industrial and commercial importers and reform the SRO regime.\nThe | government will do well to focus less on just cutting imports via high import duties which, by the way, have little impact on containing the widening CAD. They suggest forgoing exports led growth through an overly optimistic forecast and policy framework centered on Gwadar port that may just not come through on investment flows is foolish economic hubris. They suggest forgoing exports led growth through an overly optimistic forecast and policy framework centered on Gwadar port that may | 20.5 |

Table 15: Texts generated by the Adaptive Soft-Watermark on C4.

| Prompt | Adaptive Soft-Watermark | z-score |
|---|---|---|
| [...]of polyester, but there are so many variables (shipping, manufacturing, etc) that I don't think it's really confirmable which is greener.\nOf course, not every child has a parent who can, or has time to, cut a pattern, buy fabric, cut fabric, fit, and sew a costume. I definitely understand the appeal of just being able to buy one at Target and being done. Stores also have trend-based cost | umes that are more realistic than the homemade kind, which, if you're not a crafty person, may be more appealing.\nI think the new Target commercial is well-made and funny, but it also makes me a little sad. I'm glad my mom had the time and resources to make my costumes, and I'm sure the new ones will look super cute on the kids who buy them. But I'm also a little | 6.1 |
| [...]other safety equipment, setting a record for "most buildings climbed unassisted" according to Guinness World Records.\nOn his website, Robert says, "Climbing is my passion, my philosophy of life. Although I suffer from vertigo, although my accidents left me disabled up to 66%, I have become the best solo climber."\nHe encourages others by saying, "We set ourselves limits, but we are all strong enough to aim higher, to | the short term.\n"We are immediately announcing plans to exit non-core markets and geographies, raise up to a further £125m in the next 12 months to ensure we have sufficient liquidity to meet our financial obligations and to ensure we are well funded to take advantage of the new opportunities that will arise as we reposition the business.\n"We are also announcing plans to carry out a comprehensive review of the business | 10.1 |
| [...]the strategic priorities we set out in our 2016 results announcement in March, average net borrowing has increased above the level we expected, which means that we will no longer be able to meet our target of reducing leverage for the full year.\n"We have therefore concluded that we must take immediate action to accelerate the reduction in average net borrowing and are announcing a comprehensive programme of measures to address that, aimed at generating significant cashflow in | ra, Bodack said, "I want to hope and change, and I think Toomey can bring change."\nLiz Peterson, 28, a registered independent, said her vote this year was driven by her desire to ensure that her children have a better future. She voted for Democrat Bob Casey for U.S. Senate and Republican Charlie Dent for U.S. Congress.\nLiz Peterson said her vote this year was driven by her desire to | 11.1 |
| [...]made up his mind this election season by asking himself whether he was better off than he was in 2008. He decided he was not.\nJeff Bodack, 54, a registered Democrat who has been voting for Republicans since the Ronald Reagan-era, said he wants Toomey in Washington because he wants "change," and said his vote felt "urgent."\nHarking to Obama\'s 2008 campaign mant | push beyond ourselves."\nNot everyone is a fan of Robert, however. Some have criticized him for invading private property and putting himself and others at risk.\nNot much is known about Robert, other than he hails from France and has been doing this for over two decades. He has added seven new buildings to his climbing list this year alone.\nNot much is known about Robert, other than he hails from France and has been doing this for over two decades. | 19.2 |
| [...]of the trade regime.\nThey further stressed trade policy decisions should not be driven by short-term fiscal considerations and should not be made by the FBR. Instead, they proposed such decisions be made by an independent policy board or the ministry of commerce and aim to reduce anti-export bias, cost of production, tariffs on imported raw materials and intermediate goods, while eliminating the distinction between industrial and commercial importers and reform the SRO regime.\nThe | government has introduced a series of measures to spur investment in the country, including the new investment policy that aims to attract $10 billion in foreign investment over the next five years.\nThe government has introduced a series of measures to spur investment in the country, including the new investment policy that aims to attract $10 billion in foreign investment over the next five years.\nThe government has introduced a series of measures to spur investment in the | 15.2 |

