# OpenReview forum: "Necessary and Sufficient Watermark for Large Language Models"
_ICLR.cc/2024/Conference — Submitted to ICLR 2024_

### Official Review · Reviewer_kQfc · 2023-10-22

**Soundness:** 2 fair
**Presentation:** 3 good
**Contribution:** 1 poor
**Rating:** 3
**Confidence:** 4

**Summary:**

The manuscript delineates the introduction of a Necessary and Sufficient (NS) Watermark, advancing the optimization constraints integral to the formerly introduced Hard watermark. Empirical scrutiny validates that the new watermark upholds the integrity of text quality, ensuring no discernible degradation.

**Strengths:**

The comprehensiveness of the experiments underlines the efficacy of the NS algorithm proposed.

**Weaknesses:**

Major Concerns:

Predominantly, there is apprehension regarding the foundational framework of the NS watermark, which is an extension of the Hard watermark as discussed in [1]. The predecessor watermark manifests suboptimal performance particularly in scenarios involving low entropy sentences, characterized by an initial sequence of tokens that heavily influences subsequent ones. This limitation is addressed superficially through the hard red list rule, simply by precluding the language model from generating such sequences. Should the NS watermark inherit this rule, it would ostensibly undermine the integrity of low entropy sentence generation, thereby standing in contradiction to the authors' assurance of preserved text quality.

The second concern pertains to the optimization schema posited in the study. The assertion that the Hard watermark adheres to optimization problem (3) is contested by the operational reality of Large Language Models (LLMs), which employ an autoregressive mechanism, incorporating stochastic elements such as temperature adjustments, beam searches, and top-k selections, rather than strictly maximizing token probability in alignment with the language models. This discrepancy casts doubt on the theoretical soundness of the NS watermark's conceptual underpinnings.

Minor Concerns:

There is an implication that the robustness of the watermark is compromised to enhance text quality. Despite the provision to manipulate parameters $\gamma$ and $\beta$ within the optimization constraints to bolster robustness, the resilience of the proposed scheme does not rival that of [1].

Discrepancies are evident in Table 1, showcasing a discernible decline in the BLEU score when comparing NS watermark implementations with those devoid of watermarks. This observation counters the assertion of non-degraded text quality through NS watermarking. The absence of a theoretical exposition on the NS watermark’s impact on text quality further weakens the claim, indicating that the empirical data presented may not suffice to substantiate the purported benefits of NS watermarking.

[1] A watermark for large language models. Kirchenbauer et al.

**Questions:**

It is surprising that Soft-Watermark [1] has such a low BLEU score in the machine translation task. According the result reported in [1], the perplexity will not degrade too much when beam search is applied. This raises inquiries regarding whether the optimal experimental conditions were enlisted for the Soft watermark in the experiment.

---

> ### Author Response · Authors · 2023-11-22
> **Reply to kQfc**
>
> > Predominantly, there is apprehension regarding the foundational framework of the NS watermark, which is an extension of the Hard watermark as discussed in [1]. The predecessor watermark manifests suboptimal performance particularly in scenarios involving low entropy sentences, characterized by an initial sequence of tokens that heavily influences subsequent ones. [...]
>
> We assume that the reviewer criticizes the NS-Watermark from the point that the prior works, such as [2], showed that inserting watermarks into low-entropy texts may degrade the text quality.
> However, our work studied the watermarking methods from different points of view
> and focused on the minimum constraints for watermarked texts to be distinguishable.
> Specifically, we found that the minimum constraints for watermarked texts to be distinguishable can be relaxed when the generated texts are long (see Eq. (4))
> and proposed a novel watermarking method that suppresses text quality degradation by imposing a minimum constraint on the generated text based on the text length.
> The NS-Watermark is the first watermarking method that reduces text quality degradation by focusing on the length of generated texts.
> We agree that considering the effect of low/high-entropy texts is important, but the discussion on low/high-entropy texts and the minimum constraints of watermarks discussed in our study are different and orthogonal.
>
> > The second concern pertains to the optimization schema posited in the study. The assertion that the Hard watermark adheres to optimization problem (3) is contested by the operational reality of Large Language Models (LLMs), which employ an autoregressive mechanism, incorporating stochastic elements such as temperature adjustments, beam searches, and top-k selections, rather than strictly maximizing token probability in alignment with the language models. [...]
>
> We only showed the results when the beam search is used for decoding, but it is easy to extend Alg. 1 to other decoding methods (e.g., top-k sampling) as in the Soft-Watermark.
>
> > There is an implication that the robustness of the watermark is compromised to enhance text quality. Despite the provision to manipulate parameters $\gamma$ and $\beta$  within the optimization constraints to bolster robustness, the resilience of the proposed scheme does not rival that of [1].
>
> Generally speaking, there exists a trade-off between text quality and the robustness of watermarks against post-editing.
> The Soft-Watermark controls this trade-off by tuning the hyperparameters,
> but as shown in our experiments, the Soft-Watermark cannot change the ratio of green words according to the text length,
> inserts more than the required number of green words in the generated texts, and degrade the text quality more than unnecessary.
> In contrast, when the level of robustness to be achieved is given by $\beta$,
> the NS-Watermark can minimize the degradation of text quality under the conditions that it achieves the given level of robustness,
> which is the main advantage of the NS-Watermark over the Soft-Watermark.
>
> >  Discrepancies are evident in Table 1, showcasing a discernible decline in the BLEU score when comparing NS watermark implementations with those devoid of watermarks. This observation counters the assertion of non-degraded text quality through NS watermarking. [...]
>
> We will replace the phrase "watermarking methods without degrading the text quality'' with "watermarking methods that minimize the degradation of text quality'' in the revised manuscript.
>
> > It is surprising that Soft-Watermark [1] has such a low BLEU score in the machine translation task. According the result reported in [1], the perplexity will not degrade too much when beam search is applied. [...]}
>
> One of the reasons why the quality of texts generated by the Soft-Watermark shown in our experiments is much worse than that reported in [1] is that the experimental settings are different.
> [1] showed the results of generated texts of length $200 \pm 5 $ (see page 6, Fig. 2, and Table 2 in [1]), whereas we provide the results of all generated texts, which contain both long and short texts.
>
> The main drawback of the Soft-Watermark is that the Soft-Watermark cannot change the ratio of green words.
> As shown in Fig. 3, the number of green words in texts generated by the Soft-Watermark increases linearly for the text length, and the Soft-Watermark inserts too many green words when the generated texts are long and degrades text quality more than necessary.
>
> For these reasons, the results of the Soft-Watermark shown in our experiments are much worse than those shown in [1], which motivates us to propose the NS-Watermark that can change the ratio of green words according to text length in this study.
>
>
> ### Reference
> [1] Kirchenbauer et al., ICML 2023, A Watermark for Large Language Models
>
> [2] Kuditipudi et. al., arXiv 2023, Robust Distortion-free Watermarks for Language Models

---

### Official Review · Reviewer_yeCE · 2023-11-01

**Soundness:** 2 fair
**Presentation:** 3 good
**Contribution:** 3 good
**Rating:** 5
**Confidence:** 3

**Summary:**

Recent work [1] has successfully detected texts generated by LLM through the injection of watermarks, however, this method significantly degraded the text quality. In this paper, the authors propose a novel method for watermarking generated texts, termed the “Necessary and Sufficient Watermark” (NS-Watermark). From the observations that prior work [1] overly constrained the text generation, especially on long texts, this paper relaxes the constraints such that the text quality and detection accuracy are ensured. The authors formulate the NS-Watermark as a constrained optimization problem and introduce an efficient algorithm to solve it. The empirical results demonstrate the effectiveness of the proposed method.

**Strengths:**

- This work models the watermarking problem as a constrained optimization problem and then solve it by combining dynamic programming and beam search. The authors also proposed an approximation method to reduce the complexity.
- The proposed method demonstrates superior performance compared to the baselines
- The paper is effectively structured and exhibits clear and concise writing

**Weaknesses:**

- Since NS-Watermark requires solving constrained optimization, although the authors did mention the running time in Appendix D.1, how expensive it is against baselines are missing.
- The proposed algorithms depend on the beam search. However, in the experiment, the authors only use one beam size (k = 1).
- It is not clear to me how NS-Watermark robust to attacks. For example, NS-Watermark might be removed by simply adding a list of red words?

**Questions:**

1. Why the authors only select a set of small $\gamma$? What happens with larger $\gamma$ ( > 0.1) since in [1], they measured the z-score with values of $\gamma$ varies from 0.1 to 0.9. Whether or not it is because lower $\gamma$s enhance the model performance?. Based on equation 2, If $\gamma$ is small, the z-score will significantly increase. Then, Soft-Watermark with high $\delta$ can produce high z-score without sacrificing the text quality? Is it still true to keep the same Z threshold? Please correct me if I misunderstood.
2. How does NS-Watermark compare to Soft-watermark in terms of inference time? In equation 7, the approximated NS-watermark need to generate the text without watermark first, and then solve the constrain optimization problem. Besides, the beam size $k$ and $\alpha$ significantly affect the running time.
3. In the case of robustness, making WS-Watermark more robust with post-editing decreases the performance of model substantially. For example, in figure 5a, when $\beta$ ≥ 0.1 the PPL ≥ 3 which is higher than Adaptive Soft-watermark in Table 2. Similar behaviors with Translation task in figure 5b.

---

> ### Author Response · Authors · 2023-11-17
> **Reply to yeCE**
>
> > Why the authors only select a set of small $\gamma$? What happens with larger $\gamma$ ( $> 0.1$) since in [1], they measured the z-score with values of $\gamma$ varies from 0.1 to 0.9. Whether or not it is because lower $\gamma$s enhance the model performance?. Based on equation 2, If $\gamma$ is small, the z-score will significantly increase. Then, Soft-Watermark with high $\delta$ can produce high z-score without sacrificing the text quality? Is it still true to keep the same Z threshold? Please correct me if I misunderstood.
>
> One of the reasons why $\gamma$ tuned by grid search is different from the one in [1] is that the experimental settings are different.
> [1] showed the results of generated texts of length $200 \pm 5 $ (see page 6, Fig. 2, and Table 2 in [1]), whereas we provide the results of all generated texts, which contain both long and short texts.
> When $\gamma$ is large, the z-scores short texts do not exceed the thresholds even if all words contained texts are green words (see Eq. (4)).
> Specifically, texts with lengths shorter than $\tfrac{\gamma Z^2}{1 - \gamma}+1$ cannot exceed the threshold $Z$ even if all words are green.
> Thus, $\gamma$ selected in our experiments is smaller than the one in [1], especially in machine translation tasks where generated texts are relatively short (see Figs. 2 and 3).
>
> In our experiments, the hyperparameters of the Soft-Watermark are tuned by the grid search.
> Then, the Soft-Watermark achieved the best BLUE scores when $(\gamma, \delta) = (0.01, 8)$ in machine translation tasks
> and $(\gamma, \delta) = (0.1, 6)$ in natural language generation tasks.
> Thus, by extending the search space of $\gamma$ in the grid search,
> the PPL scores in Table 2 may be increased, but the BLUE scores in Table 1 would not.
>
> > How does NS-Watermark compare to Soft-watermark in terms of inference time? In equation 7, the approximated NS-watermark need to generate the text without watermark first, and then solve the constrain optimization problem. Besides, the beam size $k$ and $\alpha$ significantly affect the running time.}
>
> The time complexity of the Soft-Watermark is $\mathcal{O}(k T_{\text{max}})$, whereas the time complexity of the NS-Watermark is $\mathcal{O}(\alpha k T_{\text{max}})$,
> where $k$ is the beam size, $T_{\text{max}}$ is the maximum number of words, and $\alpha$ is the hyperparameter for the NS-Watermark.
>
> We evaluated the running time of the Soft-Watermark, adding the results in Fig. 6.
> The results are consistent with the time complexity discussed above.
> Then, as the review mentioned, the running time of the NS-Watermark with $\alpha=1$ is approximately twice slower than the one of the Soft-Watermark
> because the NS-Watermark needs to generate texts without watermarks before generating texts with watermarks.
>
> > In the case of robustness, making WS-Watermark more robust with post-editing decreases the performance of model substantially. For example, in figure 5a, when $\beta$
>  $\geq$ 0.1 the PPL $\geq$ 3 which is higher than Adaptive Soft-watermark in Table 2. Similar behaviors with Translation task in figure 5b.
>
> Table 2 shows the results when the hyperparameters for all methods are tuned to achieve the best PPL under the condition that the false negative rate is less than $5 \%$,
> and Fig. 5 shows the results when the NS-Watermark generates texts under the condition that it achieves the level of robustness determined by the hyperparameter $\beta$.
> Thus, we cannot directly compare the results in Table 2 and Fig. 5b.
> If we tune the hyperparameters of the Adaptive Soft-Watermark as in Fig. 5 (i.e., the Adaptive Soft-Watermark achieves the false positive rate of less than $5\%$ and achieves the level of robustness determined by $\beta$),
> the PPL scores of the Adaptive Soft-Watermark may decrease as in the results of the NS-Watermark in Fig. 5.
>
> ### Reference
> [1] A watermark for large language models. Kirchenbauer et al., ICML 2023

---

### Official Review · Reviewer_V7Qr · 2023-11-02

**Soundness:** 4 excellent
**Presentation:** 3 good
**Contribution:** 4 excellent
**Rating:** 5
**Confidence:** 3

**Summary:**

In this paper, the authors propose a novel method to insert watermarks into generated text without compromising text quality. Specifically, the authors formulate the watermarking injection problem as an constrained optimization and provide a linear-time solution. The authors evaluate the effectiveness of the proposed method on machine translation and natural language generation tasks.

**Strengths:**

1.  The proposed method can preserve the quality of the generated text at a certain level.

2. The authors provide an approximation solution with linear time complexity

3. The authors conduct a series experiments to evaluate the effectiveness in terms of the text quality, the detection accuracy, and the sensitivity towards the hyper-parameters.

**Weaknesses:**

1. Regarding the methodology:

1.1 There needs to be a more in-depth discussion about the comparison between the proposed naive method and the linear-time approximation method. Since the linear-time method is an approximation method, it would be valuable to understand what it sacrifices in order to obtain the linear time complexity.

1.2 The figure 1's illustration is not very clear.  What's the difference between the two sub-figures in figure 1(a)?

2. Regarding the experiment:

2.1 Lack the comparison between the proposed naive method and the linear-time method.

2.2 My major concern: the experiment about the robustness to post-editing is not convincing. The authors only show the relation between the text quality and the hyper-parameter controlling the robustness. I expect to see is the robustness under real attacking methods. The included attack methods could refer to the three types of attacks adopted in Section 7 of the paper [1]. Without seeing such experimental results under attacking, it's hard to be convinced about the robustness of the proposed method.

[1]"A Watermark for Large language Models" by John Kirchenbauer*, Jonas Geiping*, Yuxin Wen, Jonathan Katz, Ian Miers, Tom Goldstein

**Questions:**

See the content in weakness.

---

> ### Author Response · Authors · 2023-11-17
> **Reply to V7Qr**
>
> > There needs to be a more in-depth discussion about the comparison between the proposed naive method and the linear-time approximation method. [...]
>
> See the response below.
>
> > The figure 1's illustration is not very clear. What's the difference between the two sub-figures in figure 1(a)?
>
> Fig. 1(a) is used to explain the idea described in the last paragraph in Sec. 3.2.
> Naively, the table $\mathbf{T}[t][g]$ stores the texts of length $t$ containing $g$ green words. Then, we need to fill $\mathbf{T}[t][g]$ for all
>
> $$(t,g) \in \\{ (t,g) \in \mathbb{N}^2 \mid 1 \leq t \leq T_{\text{max}}, 0 \leq g \leq t-1 \\}.$$
>
> That is, we need to fill the areas colored in blue and light blue in the right figure in Fig. 1(a).
> However, if $G_{\text{max}}$ green words appear after generating $t$ words, we do not need to count the number of green words in the remaining text because the constraint in Eq. (4) is guaranteed to be satisfied.
> To utilize this idea, we modify the table $\mathbf{T}$ such that $\mathbf{T}[t][G_{\text{max}}]$ stores the texts of length $t$ containing \textbf{at least} $G_{\text{max}}$ green words.
> After this modification, we only need to fill the table $\mathbf{T}[t][g]$ for all
> $$(t,g) \in \\{ (t,g) \in \mathbb{N}^2 \mid 1 \leq t \leq T_{\text{max}}, 0 \leq g \leq \min\\{ t-1, G_{\text{max}} \\} \\},$$
>  which is the area colored in the left figure in Fig. 1(a).
> Compared to the right figure in Fig. 1(a) with the left figure, the area in the table $\mathbf{T}$ we need to fill is shrunk. Then, the time complexity can be reduced to $\mathcal{O}(\gamma k T_{\text{max}}^2)$ from $\mathcal{O}(k T_{\text{max}}^2)$.
>
> >  Lack the comparison between the proposed naive method and the linear-time method.
>
> We compared the naive and linear-time methods in the validation data of WMT'16 En$\rightarrow$De.
> We denote the naive method by the NS-Watermark with $\alpha=\infty$.
> The results indicate that when $\gamma$ is large, the BLEU scores increase with $\alpha$,
> but when $\gamma$ is small, the BLEU scores are almost the same,
> and the results of the naive and linear-time methods are almost the same.
> Fortunately, because the NS-Watermark achieved the best BLEU score when $\gamma$ was small,
> the linear-time method can reduce the time complexity with negligible text quality degradation in practice.
> We are currently conducting the experiments with $\gamma=0.1$. We will add these experimental results in a revised version.
>
> | $\gamma$                  | $\alpha$ | BLUE   |
> |-------------|----------|--------|
> | $0.01$   | $1$      | $20.9$ |
> | $0.01$                           | $2$      | $23.1$ |
> | $0.01$                          | $3$      | $24.2$ |
> | $0.01$                           | $4$      | $26.0$ |
> | $0.01$                           | $5$      | $26.1$ |
> | $0.01$                          | $\infty$ | $26.1$ |
> | $0.001$  | $1$      | $31.5$ |
> | $0.001$                           | $2$      | $32.4$ |
> | $0.001$                           | $3$      | $32.4$ |
> | $0.001$                           | $4$      | $32.4$ |
> | $0.001$                           | $5$      | $32.4$ |
> | $0.001$                           | $\infty$ | $32.4$ |
> | $0.0001$ | $1$      | $32.6$ |
> | $0.0001$                          | $2$      | $32.6$ |
> | $0.0001$                         | $3$      | $32.6$ |
> | $0.0001$                          | $4$      | $32.6$ |
> | $0.0001$                          | $5$      | $32.6$ |
> | $0.0001$                          | $\infty$ | $32.6$ |
>
> >  the experiment about the robustness to post-editing is not convincing. The authors only show the relation between the text quality and the hyper-parameter controlling the robustness. I expect to see is the robustness under real attacking methods. [...]
>
> The main advantage of the NS-Watermark over the Soft-Watermark is that
> when the level of robustness to be achieved is given by the hyperparameter $\beta$, the NS-Watermark can minimize the degradation of text quality under the conditions that it achieves the given level of robustness.
> As shown in our experiments, the Soft-Watermark inserts more than the required number of green words in the generated texts and degrades the text quality more than necessary, even if its hyperparameters are tuned.
>
> The hyperparameter $\beta$ can be intuitively interpreted
> and ensures that up to $50\beta \\%$ of the words can be replaced without removing the NS-Watermark.
> Then, how watermarks should be robust to post-editing depends on the application.
> Thus, we leave it to future work to experimentally evaluate the robustness using the attacking methods that the reviewer mentioned.

---

> > ### Comment · Reviewer_V7Qr · 2023-11-22
> >
> > Thank you for your response. As no additional experiments against real attacking methods, I will keep my score.

---

### Official Review · Reviewer_rYag · 2023-11-03

**Soundness:** 3 good
**Presentation:** 2 fair
**Contribution:** 3 good
**Rating:** 5
**Confidence:** 4

**Summary:**

The paper proposes a new method, the Necessary and Sufficient Watermark (NS-Watermark) for watermarking texts generated by large language models (LLMs). The authors argue that the use of LLMs can be misused for malicious purposes, hence the need for an efficient watermarking method to distinguish between human and LLM-generated texts. The NS-Watermark applies minimum constraints to generated texts, maintaining text quality while ensuring accurate detection. The authors propose this method in response to existing watermarking methods that degrade the quality of generated texts. The paper demonstrates that the NS-Watermark outperforms existing methods in distinguishing machine-generated text and maintaining text quality, especially in machine translation tasks.

**Strengths:**

(1) The paper addresses a significant issue in the field of natural language processing, specifically in mitigating risks associated with the malicious use of large language models.
(2) The authors propose a novel method, the Necessary and Sufficient Watermark (NS-Watermark), which is an innovative solution to the problem at hand.
(3) The paper provides a comprehensive analysis of the proposed method, including a well-structured theoretical analysis and practical implementation details.

**Weaknesses:**

(1) It is overclaimed that the text quality is unaffected, compared to the no watermarked model. There is some quality drop compared to the unwatermarked model, but not too much. You can claim the text quality is better than Soft-Watermark. So using the phrase "without degrading the text quality" is not fully accurate.

(2) The proposed model explores a less conservative region of z-scores compared to Soft-Watermark. Soft-Watermark's conservative approach provides robustness against attacks. And the NS-Watermark is much slower in terms of decoding speed.


(3) Since watermarking is designed to distinguish watermarked text and unwatermarked text, it makes more sense to consider human text and unwatermarked AI-generated text together as negative samples.

**Questions:**

Since the watermarked tokens are dynamically added, if the adversary truncates the text and only provides part of the generated text, will this affect the detection performance?

---

> ### Author Response · Authors · 2023-11-17
> **Reply to rYag**
>
> > It is overclaimed that the text quality is unaffected, compared to the no watermarked model. [...] So using the phrase "without degrading the text quality" is not fully accurate.
>
> We will replace the phrase "watermarking methods without degrading the text quality'' with "watermarking methods that minimize the degradation of text quality'' in the revised manuscript.
>
> > The proposed model explores a less conservative region of z-scores compared to Soft-Watermark. Soft-Watermark's conservative approach provides robustness against attacks. [...]
>
> Generally, there is a trade-off between text quality degradation and robustness against post-editing.
> Thus, the main advantage of the NS-Watermark over the Soft-Watermark is that
> when the level of robustness to be achieved is given, the NS-Watermark can minimize the degradation of text quality under the conditions that it achieves the given level of robustness,
> whereas the Soft-Watermark cannot and inserts more than the required number of green words in the generated texts.
>
> More specifically, by using hyperparameters $\beta$, the NS-Watermark can insert the minimum number of green words to achieve the level of robustness determined by $\beta$.
> In contrast, even if the hyperparameters $\gamma$ and $\delta$ are tuned, the Soft-Watermark cannot control the number of green words contained in the generated texts
> and degrades the text quality more than necessary, as shown in Fig. 3 and Th. 1.
>
> > And the NS-Watermark is much slower in terms of decoding speed.
>
> We add the experiments, comparing the running time of the NS-Watermark and Soft-Watermark in Fig. 6.
> The results indicate that the running time of the NS-Watermark is at most twice slower than that of the Soft-Watermark when $\alpha=1$.
> Note that the NS-Watermark achieved the best BLUE score when $(\gamma, \alpha)=(0.0001, 1)$ (see Fig. 4).
>
> > Since watermarking is designed to distinguish watermarked text and unwatermarked text, it makes more sense to consider human text and unwatermarked AI-generated text together as negative samples.
>
> As the reviewer mentioned, LLM-generated texts without watermarks are also considered negative samples in the formulations of the Hard/Soft-Watermark and NS-Watermark.
> However, for the simplicity of explanation, we ignore LLM-generated texts without watermarks, as in the prior work [1].
>
> > Since the watermarked tokens are dynamically added, if the adversary truncates the text and only provides part of the generated text, will this affect the detection performance?
>
> Truncation is also regarded as post-editing.
> Thus, we can make the NS-Watermark robust to the truncation by using positive $\beta$.
> When we set $\beta$ to zero, the truncation can remove the NS-Watermark because the number of green words inserted by the NS-Watermark is minimal and depends on the length of generated texts (see Eq. (4)).
>
> ### Reference
> [1] A watermark for large language models. Kirchenbauer et al., ICML 2023

---

### Meta-Review · Area_Chair_pwNs · 2023-12-06

**Metareview:**

The paper introduces NS-Watermark, a new watermarking technique for LLMs. The primary concerns revolve around overstatements about text quality, insufficient robustness against real-world attacks, lack of clarity in methodology, and issues with the foundational framework of the NS watermark. Additionally, the paper doesn't convincingly address the efficiency and practicality of the proposed method compared to existing techniques like Soft-Watermark.

**Justification For Why Not Higher Score:**

The paper's claims about the text quality of NS-Watermark aren't fully on point. It actually does drop a bit compared to unwatermarked models. Plus, the robustness against real attacks isn't convincing – they didn't test it against tough, real-world attack methods, which is a major issues. There's also a bit of a unclearity around their methods, like how the linear-time method trades off against the naive one. And, the whole foundation of NS-Watermark seems to have some inherited flaws from older methods, which they don't really clear up.

**Justification For Why Not Lower Score:**

NA

---

### Decision · Program_Chairs · 2024-01-16

Reject